# Differential Expression of the Tetraspanin CD9 in Normal and Leukemic Stem Cells

**DOI:** 10.3390/biology10040312

**Published:** 2021-04-08

**Authors:** Rachid Lahlil, Maurice Scrofani, Anne Aries, Philippe Hénon, Bernard Drénou

**Affiliations:** 1Institut de Recherche en Hématologie et Transplantation (IRHT), Hôpital du Hasenrain, 87 Avenue d’Altkirch, 68100 Mulhouse, France; scofanim@ghrmsa.fr (M.S.); ariesa@ghrmsa.fr (A.A.); DRENOUB@ghrmsa.fr (B.D.); 2CellProthera, 12, rue du Parc, 68100 Mulhouse, France; phenon@cellprothera.com; 3Laboratoire d’Hématologie, Groupe Hospitalier de la Région de Mulhouse Sud-Alsace, Hôpital E. Muller, 20 Avenue de Dr Laennec, 68100 Mulhouse, France

**Keywords:** VSELs, human very small embryonic-like stem cells, umbilical cord blood, CD9, CD34^+^/CD133^+^/CXCR4^+^ cells, leukemia

## Abstract

**Simple Summary:**

Before their use in regenerative medicine, stem cells need to be expanded to obtain sufficient cells for the efficient reparation of the injured tissues. This expansion must not affect their integrity. Regarding the role played by different receptors, we observed that, during their expansion, the number of promising pluripotent stem cells found in adult tissues, i.e., very small embryonic-like stem cells (VSELs), which express the CD9 receptor, decreased. This is due to their higher mortality rate compared to that of those not expressing CD9, which can lead to low regenerative efficiency for injured tissues. Interestingly, this could be overcome by the addition of a specific growth factor, allowing the re-establishment of their function. Finally, we found that the expression of this receptor is also deregulated in cells phenotypically identical to VSELs isolated from leukemic patients, which attests to the instability of its expression and may explain disease progression.

**Abstract:**

CD9 plays a crucial role in cellular growth, mobility, and signal transduction, as well as in hematological malignancy. In myeloid neoplasms, CD9 is involved in the altered interactions between leukemic and stromal cells. However, apart from its role in CD34^+^ progenitors and myeloid and megakaryocytic differentiation, its function in normal and leukemic pluripotent cells has not yet been determined. Very small embryonic-like stem cells (VSELs) are promising pluripotent stem cells found in adult tissues that can be developed for safe and efficient regenerative medicine. VSELs express different surface receptors of the highest importance in cell functioning, including CD9, and can be effectively mobilized after organ injury or in leukemic patients. In the present study, we observed that CD9 is among the most expressed receptors in VSELs under steady-state conditions; however, once the VSELs are expanded, CD9^+^ VSELs decrease and are more apoptotic. CD9^–^ VSELs had no proliferative improvement in vitro compared to those that were CD9^+^. Interestingly, the addition of SDF-1 induced CD9 expression on the surface of VSELs, as observed by flow cytometry, and improved their migration. In addition, we observed, in the phenotypically identical VSELs present in the peripheral blood of patients with myeloproliferative neoplasms, compared to healthy subjects, a significantly higher number of CD9^+^ cells. However, in their hematopoietic stem cell (HSC) counterparts, the expression remained comparable. These results indicate that, likewise, in progenitors and mature cells, CD9 may play an important function in normal and malignant VSELs. This could explain the refractoriness observed by some groups of expanded stem cells to repairing efficiently damaged tissue when used as a source in cell therapies. Understanding the function of the CD9 receptor in normal and malignant CD34^+^ and VSELs, along with its relationship with the CXCR4/SDF-1 pathway, will enable advances in the field of adult pluripotent cell usage in regenerative medicine and in their role in leukemia.

## 1. Introduction

Stem cell research has revealed the ability of some CD34^+^ stem cell populations present in blood or bone marrow to produce different specialized mature cells, able to substitute malignant leukemic cells and regenerate injured regions within an organ [1,2]. Such discoveries highlight their potential use in the handling of many diseases, such as myocardial infarction, diabetes, or Alzheimer’s, for which it is difficult to find effective conventional treatments at present [3,4]. The current study and understanding of stem cell expansion, mobilization, and differentiation remain a major challenge in the search for cellular sources ready-to-use for regenerative medicine [1]. As pluripotent stem cells, very small embryonic-like stem cells (VSELs) are a subpopulation of more immature CD34^+^ cells [5,6], representing small and quiescent cells found in many tissues and having significant differentiation capabilities to many specific lineages [7,8,9,10]. We previously demonstrated that they were present in human peripheral blood taken from voluntary donors of different ages [11]. VSELs constitute a lifelong supply of pluripotent cells capable of being mobilized from bone marrow to an injured organ, and in patients with leukemia, they represent a real hope for regenerative medicine [12,13,14]. Nevertheless, their correct expansion in order to have an appropriate amount of identical autologous cells remains essential before their application in regenerative medicine [15,16]. VSELs express different receptor types; they are lineage-negative, CD34^+^ CD45^−^, and express CD133 and/or CXCR4 receptors.

As a member of the tetraspanin superfamily, CD9 is involved in the regulation of various physiological processes, such as cell motility, adhesion, and fusion [17,18]. In addition, it appears that the expression of this receptor plays a key role in the implantation of umbilical cord blood stem cells into bone marrow [19]. Other studies have shown that CD34^+^ CD9^−^ stem cells have a significantly lower engraftment capacity than those that are positive for the CD9 receptor, assigning it specific functions in the homing of stem cells [20]. Interestingly, the expression of CD9 is regulated by the CXCR4/SDF-1 axis through different signaling molecules, including those of the MAPK and AKT pathways [19]. On the contrary, CD9 appears to be a significant element in leukemic cell development, and its expression is upregulated in CD34^+^ cells from patients affected by myeloid neoplasms [21] and in multiple myeloma cells [18]. In primary myelofibrosis, CD9 is involved in the altered interactions between megakaryocytes and bone marrow mesenchymal stromal cells and participates in alterations affecting the survival, differentiation, and stromal cell-derived factor CXCL12 (SDF-1) with the CXC chemokine receptor 4 (CXCR4)-mediated migration of megakaryocytes, contributing to the occurrence of dysmegakaryopoiesis [22]. Finally, CD9 expression can predict a TEL/AML1 rearrangement in acute lymphoblastic leukemia [23].

In the present study, we dissected our previous observation through a global study of mRNA expression, indicating that under steady-state conditions, the CD9 receptor is intensely overexpressed in VSELs compared to control cells [15]. However, this strong presence appears to decrease after a few days of their expansion, accompanied by an increase in their apoptosis without affecting their proliferation, indicating that the CD9 receptor expression is regulated in VSELs. The in vitro migration of CD9^-^VSELs seems to be reduced. We then examined whether the addition of SDF-1 in the culture medium during the expansion of VSELs can overcome these effects by stimulating the expression of the CD9 receptor or not. Interestingly, our results indicated that the percentage of VSELs that are Lin^–^CD34^+^CD45^−^ CD133^+^ and/or CXCR4^+^ positive for the CD9 receptor increases significantly in a dose-dependent manner with the addition of SDF-1. This demonstrates the importance of the presence of SDF-1 in the expansion medium to support CD9 expression in VSELs. Finally, we also found that CD9 expression is deregulated in cells phenotypically identical to VSELs isolated from leukemic patients. Compared to control VSELs and hematopoietic stem cells (HSCs) from healthy subjects, CD9 is overexpressed in VSELs “like” those isolated from the peripheral blood of patients affected by myeloproliferative neoplasms (MPNs). Since tetraspanin proteins play an important role in the communication between leukemic cells and the bone marrow microenvironment and are of significant interest in cell trafficking and adhesion to bone marrow [22,23,24], deregulated expression of CD9 in patients with leukemia may lead to and explain disease progression.

## 2. Materials and Methods

### 2.1. Isolation of VSELs and Flow Cytometry

Umbilical cord blood (UCB) samples were obtained from healthy persons, with written informed consent and with the approval of local human subject research ethics boards (CED EFS, Besançon, France). Briefly, human UCB mononuclear cells were collected by centrifugation after reduction of red blood cells by Gelofusine (B Braun) treatment, as described previously [11], followed by an additional red blood cell lysis with ammonium chloride lysis buffer (STEMCELL Technologies, Grenoble, France). Cells were then incubated with a cocktail of lineage-specific antibodies from an EasySep™ progenitor cell enrichment kit with platelet depletion and human CD45 depletion kits (STEMCELL Technologies) for immuno-magnetic negative selection of Lin^–^CD45^−^ cells using an EasySep™ magnet (STEMCELL Technologies). Purified VSELs were expanded in stemSpan media (STEMCELL Technologies) containing 50 ng/mL of TPO, 10 ng/mL of SF, 10 ng/mL of FLT3, and 35 nM of UM171 for 12 days. Depending on the experiment, VSELs that were either expanded or not, were then stained with a mixture of lineages (Lin), associating monoclonal antibodies (MoAbs) conjugated with fluorescein isothiocyanate (FITC). At the same time, V500-conjugated CD45 (Beckman Coulter, Villepinte, France), CD34 PE clone 8G12, and a combination of allophycocyanin (APC)-conjugated MoAbs, CD133 clone AC133 (Miltenyi Biotec, Paris, France), or CD184 (CXCR4) clone 12G5 (BD Biosciences, Grenoble, France), were added for 30 min on ice. For VSEL isolation, labeled cells were washed and discriminated by flow cytometry on the basis of cell size, granularity, presence of CD34, and absence of Lin and CD45 markers. Then, depending on the VSEL population studied, the presence of CD133 and/or CXCR4 markers was gated. Cell viability was monitored by the absence of 7-AAD dye (BD Biosciences) uptake, which was added 10 min before acquisition. All flow cytometry sorting or analysis was performed using a BD ARIA III instrument (BD Biosciences). Data acquisition and analysis were conducted using BD FACSDiva software (BD Biosciences).

### 2.2. RNA-Sequencing and Data Processing

RNAseq was performed on 300 sorted VSELs expressing *Nanog* and control cells negative for *Nanog*. The mRNA was retro-transcribed on cDNA using a smarter ultra-low input RNA kit for sequencing (Clontech, Saint-Germain-en-Laye, France). Detailed procedures specific to VSEL isolation were described previously in [15]. Briefly, sample quality was assessed using Bioanalyzer RNA Nanochips (Agilent, Technologies, Les Ulis, France). Paired-end, barcoded RNA-Seq sequencing libraries were then generated using the Nextera XT DNA library preparation kit (Illumina, Evry, France) following the manufacturer’s protocols. Sequencing was performed using an Illumina HiSeq2500 with TruSeq SBS v3 chemistry on the iGE3 Genomics Platform (University of Geneva). Normalization and differential expression analysis were performed with the R/Bioconductor package edgeR v.3.10.5 for the genes annotated in the reference genome. The raw count data were filtered. We filtered out very lowly expressed genes, keeping genes that were expressed at a reasonable level (10 counts in at least two samples). The filtered data were normalized by the library size, and differentially expressed genes were estimated using the generalized linear model (GLM) approach. The fold change (FC) of the base 2 logarithm of the trimmed mean of the M-values normalization method (TMM normalized data) log_2_FC was used to rank the data from the top upregulated to the top downregulated genes, and the FDR (0.05) was used to define significantly differentially expressed genes.

### 2.3. Real-Time RT–PCR Analysis

For all the real-time RT–PCR determinations, total cellular RNA was isolated with an RNAeasy kit according to the manufacturer’s instructions (QIAGEN, Courtaboeuf, France), and cDNA was synthesized using an iScript cDNA Synthesis Kit (BIO-RAD, Marnes-la-Coquette, France). Real-time RT-PCR was carried out in triplicate with SYBR^®^ Green PCR Supermix (BIO-RAD). The mRNA content of the compared samples was normalized based on the amplification of GAPDH and/or β2-microglobulin. The oligonucleotides used for real-time RT-PCR gene amplification from human UCB are summarized in Table 1.

### 2.4. In Vitro Migration Assay

Transwell migration assays were performed by loading on the top chamber 10^5^ to 10^6^ EasySep purified CD34^+^CD45^–^ cells in StemSpan medium. SDF-1 (100 ng/mL, R&D System) was added or not added to the bottom chamber. Cells were incubated for the indicated time (6, 24, or 48 h); the migration percentage was calculated after quantification of live cells in both chambers with FACS analysis.

### 2.5. Peripheral Blood Samples from Patients and Controls

Normal donors (Etablissement Français du Sang, Strasbourg, France) and myeloproliferative neoplasm samples (GHRMSA, Mulhouse Hospital) were taken following informed consent and performed in accordance with the ethical standards of the Declaration of Helsinki. The form of leukemia diagnosis in patients was made according to the World Health Organization (WHO) classification [25] by analysis of the presence of (V617F) mutations. All patients were in a stage of myeloproliferative neoplasm disease, and all had the primary form at the time of diagnosis.

### 2.6. Statistical Analysis

The results are expressed as the mean ± standard deviation of at least three experiments. Statistical differences between patients and controls or between conditions were validated by *t*-tests, with *p* < 0.05 being considered statistically significant.

## 3. Results

### 3.1. The CD9 Receptor Expression in the VSEL Subpopulation

By reanalyzing the data previously obtained through a global study of mRNA expression by RNA sequencing, when comparing the VSELs isolated on the basis of the expression of the pluripotent-specific transcription factor *Nanog*+ vs. control cells negative for its expression [15], we observed that several receptors were significantly up- or downregulated in VSELs under steady-state conditions. Interestingly, CD9 had relatively the most induced expression in VSELs (log_2_ FC = 7.22) (Figure 1).

We then, however, determined the behavior of CD9 in VSELs by first verifying its expression by real-time PCR. We quantified the presence of CD9 mRNA in VSELs maintained for three days or those expanded 12 days and compared them to control cells (not VSELs) corresponding to the same indicated days. The results showed that on the third day of culture, the mRNAs of CD9 in VSELs were six to seven times higher than the controls (Figure 2a). However, when VSELs were maintained for 12 days in the expansion medium, its expression seems to be comparable to that of control cells (Figure 2b). The quantification of CD9 expression by flow cytometry in VSEL CD133^+^ on days 3 and 12 showed that only a few cells express this receptor—13% and 1%, respectively (Figure 2c). Meanwhile, in comparison to CD22, another receptor whose expression was also found upregulated in VSELs (Figure 1), we observed that approximately 96% and 94% remained positive on days 3 and 12, respectively. These results demonstrate that in contrast to other receptors, the level of expression of CD9 is affected during the amplification of VSELs, which would lead—under these conditions of expansion—to the failure of a cellular therapy projected via the amplified VSELs. Overall, the upregulated expression observed on the first day of their isolation in the RNAseq study seemed to decline within 3 to 12 days of their expansion, which indicates that the expression of this receptor is regulated in VSELs and that an essential factor for their maintenance is missing during their expansion. Nevertheless, because of their scarcity, using VSELs as a source in regenerative medicine requires that their number be amplified ex vivo.

### 3.2. The SDF-1 Effect on CD9 Receptor Expression in VSEL Subpopulations

Regarding the role played by CD9, as described above, using VSELs lacking this receptor in cell therapy can lead to their low regenerative efficiency and prevent them reaching the regions containing injured tissues. It is suggested that SDF-1 binding to the CXCR4 receptor improves the function of different cells by upregulating the expression of CD9 and, thus, impacting stem cell mobilization to the injured region [19,26]. In order to determine the effect of SDF-1 on CD9 receptor expression in VSELs, the purified and expanded stem cell Lin^-^CD34^+^CD45^−^ fraction enriched with VSELs were incubated for 4 h in the presence of different concentrations of SDF-1 (10, 50, and 100 ng/mL). The presence of VSELs positive for CD9 expression was then determined by flow cytometry (Figure 3a,b). Our results show that, in the presence of SDF-1, the percentage of VSELs CD34^+^CD45^−^CD133^+^CXCR4^+^ that expresses the CD9 receptor increased significantly in a dose-dependent manner. This demonstrates the importance of the presence of SDF-1 in the expansion medium to support expression of the CD9 receptor in VSELs.

### 3.3. CD9 Receptor Expression and VSELs Migration

Since SDF-1 is able to induce CD9 expression and the SDF-1/CD9 axis is important in hematopoietic stem cell mobilization [19,26], we decided to determine its effect on VSEL migration in vitro. We purified the Lin-CD34^+^CD45^–^ cells from UCB, and we measured their ability to migrate through a defined membrane by using a Boyden chamber. After 12 h of migration, 43.8% of the VSELs were found positive for CD9 in the lower chamber against 20.3% in the upper chamber (Figure 4A). On average, as shown in Figure 4B, we observed that CD9^+^ VSELs have significantly more migration capacity in vitro than CD9^–^ VSELs (11.43 ± 12.14 vs. 23.68 ± 18.02). We then sorted VSELs positive and negative for CD9 and measured their migration during 4 h in the presence or absence of SDF-1. We observed that CD9^+^ VSELs migrated better than CD9^–^ VSELs. As shown in Figure 4C, more CD9^+^ VSELs crossed the membrane to the lower chamber during the 4 h of culture, and in the presence of SDF-1, their migration was further improved. Although CD9 can only be a surrogate of VSELs cell differentiation during expansion, these results highlight the improvement in their migration capacities.

### 3.4. CD9 Receptor Expression and VSELs Survival and Proliferation

We then looked at whether the expansion of VSELs selectively affects the survival and proliferation of those negative for CD9 expression at the expense of CD9^+^. Interestingly, flow cytometry analysis of apoptotic VSELs by annexin V labeling showed that VSELs expressing the CD9 receptor are more apoptotic, with 10.6% of cells positive for annexin V versus less than 1% of CD9-negative cells (Figure 5a). Real-time RT-PCR analysis of mRNA expression representative of proliferative genes showed no differences in the expression levels of Ki67, p57, or AURKA, suggesting that CD9^−^ VSELs do not proliferate faster at the expense of CD9^+^. However, the mRNA of the *p21* gene, known as an important actor preventing cells from undergoing apoptosis, is significantly decreased in CD9^+^ VSELs, which would explain the reduction in their number during expansion (Figure 5b).

### 3.5. CD9 Receptor Expression in VSELs and Leukemia

According to several studies, it appears that CD9 expression levels on the surface of hematopoietic cells are affected during the occurrence of different types of leukemia [23,27]. In order to verify if this event also occurs in pathologic cells that are phenotypically identical to VSELs (VSELs-like), we analyzed CD9 expression in HSCs and their supposed precursors, VSELs taken from the peripheral blood of patients newly diagnosed with myeloproliferative neoplasms, in comparison to those of healthy subjects. After blood count confirmation (Table 2), the quantification of CD9 in VSELs-like cells purified from the peripheral blood cells of patients diagnosed with *JAK2* (V617F) showed that its expression was significantly increased in the leukemia patients’ cells in comparison to the control cells isolated from healthy donors (1.1 ± 1.22% vs. 0.16 ± 0.18% of lineage-negative cells, respectively) (Table 2). Interestingly, the HSCs expressing CD9 versus lineage-negative cells remain comparable between control cells (0.14 ± 0.1%) and newly diagnosed patients 0.17 ± 0.15% (Table 2). The induction of CD9 receptor expression in VSELs may therefore have a potential link with the development of some MPNs.

## 4. Discussion

The homing of stem cells to their niche is a crucial process in the success of a stem cell engraftment. When HSCs are transplanted, they are able to reach the bone marrow and graft to establish their niche. This highly regulated process may represent a broader procedure, whereby different circulating stem cells migrate to particular tissue territories. The molecular mechanisms involved in this process are still poorly understood, but experimental evidence shows the involvement of chemokines, adhesion molecules, and proteolytic enzymes [28,29].

In this study, we observed that an important actor known as playing a primordial role in these biological processes can be affected during the expansion of stem cells as VSELs. This decrease could be explained by the increase in apoptosis induced in CD9^+^ VSELs due to mechanisms that are still unknown. In addition, this is supported by the finding that p21, an important actor that several studies have shown is able to protect numerous cell types from apoptosis, transcriptionally decreases. When their number has been amplified ex vivo, the loss of CD9^+^ VSELs during their expansion can lead, as described previously for HSCs [19,20], to their low efficiency to reach regions containing damaged tissues when used as a cell source in regenerative medicine. Indeed, recent studies have shown that CD34^+^ stem cells amplified by means of different culture media containing different chemical molecules and/or growth factors have significantly lower engrafting capacities than those freshly harvested [30].

The condition and expansion medium that we used was clearly devoid of an essential factor for the maintenance of CD9 expression. Interestingly, by adding SDF-1 during the 12 days necessary for the optimal expansion of VSELs and HSCs, we observed that it can significantly overcome the loss of CD9 expression in these adult pluripotent stem cells. Regarding the importance of CD9, we can conceive and hope that the expanded cells become physiologically close—if not identical—to freshly isolated cells. Indeed, SDF-1 and its receptor, CXCR4, play an important role in the mobility and homing of stem cells [28,31]. In vitro, SDF-1 induces chemotaxis, trans-endothelial migration, and adhesion to the extracellular matrix. In addition, the homing and engraftment of CD34^+^ cells in NOD/SCID mice are greatly altered by the neutralization of CXCR4 or desensitization with high doses of SDF-1 [31]. However, we cannot exclude that other compounds are also affected, because another study showed that the CXCR4^−/−^ fetal liver cells are capable of engraftment in the bone marrow of wild-type mice (although at a lower level), suggesting that HSC homing is not exclusively controlled by the SDF-1/CXCR4 axis [32]. Although other known mechanisms remain rare, the axes centered on the integrin VLA4 (very late antigen 4) and sphingosine-1-phosphate (S1P) are the best identified [33]. The exposure of CD34^+^ cells to SDF-1 leads to an increase in the rate of their migration and engraftment in mice [19], which is in agreement with the observed increase in VSELs migration in the presence of SDF-1.

The role of CD9 in leukemia remains controversial. In acute lymphoblastic leukemia, CD9 seems to be associated with cancer stem cell properties, is involved in leukemic progression, and is related to unfavorable outcomes [34]. Conversely, in acute myeloid leukemia, CD9 is associated with a favorable outcome and could be a very relevant marker for minimal residual disease monitoring [27,35]. In our study, we focused on the expression of CD9 in myeloproliferative neoplasms—specifically those linked to the *JAK2* (V617F) mutation, such as polycythemia vera, and essential thrombocythemia. We found that the levels of CD9 expression in HSCs do not seem to be affected, unlike in cells phenotypically identical to VSELs, suggesting their potential activation and mobilization from bone marrow to peripheral blood when disease occurs. The VSELs can then support the levels of HSC pools, which, as reviewed by Yamashita et al. [36], differentiate to progenitors due to the overproduction of certain lineages of blood and the activation of emergency hematopoiesis in NMPs. Furthermore, as with normal VSELs, the SDF-1/CXCR4 pathway could also be affected, since mice deficient in SDF-1 or CXCR4 are—in addition to being severely affected during the development in the homing of stem cells in the bone marrow—deficient in the establishment of lymphopoiesis B and myelopoiesis [37].

The increasing evidence based on cellular and molecular studies has made it possible to admit that, often, the cell of origin of some leukemias could be a more primitive cell, such as VSELs, than an engaged progenitor, which lacks the potential for self-renewal. However, it is not yet known whether leukemic VSELs still have the same physiology as those identified in healthy subjects and described by several groups. This, however, constrains us to remaining aware until we prove that these interesting cells, which are not yet well-known nor studied, express a greater level of the CD9 receptor than normal VSELs isolated from healthy subjects, have a pluripotent capacity, and are able to differentiate to HSC cells and the other lineages of germ layers. Nevertheless, it is well described that VSELs are immobilized into peripheral blood after leukemic disorder, most likely in order to reinject new CSHs in patients affected by the known genetic alterations responsible for the initiation of NMPs. This raises an interesting question about their molecular state in comparison to CSHs and opens new perspectives for understanding the biology of VESL-like present in patients with blood disorders.

## 5. Conclusions

The concept that an expansion of VSELs would be sufficient for their use in regenerative medicine seems to come up against the need to verify their integrity. Indeed, this study of the presence of CD9 in normal and pathological VSELs allowed us to highlight their physiological difference to HSCs. Such a finding clearly shows the variations in the function and homeostasis of stem cells depending on the expression of an important actor such as CD9 receptor. The use of VSELs in which the expression of CD9 has been induced, or those naturally positive for this receptor, could constitute a simple and reasonable solution that could increase the regenerative efficiency of these pluripotent cells. However, these results must be confirmed through experiences of loss and gain of function of CD9 to show its direct implication in these biological events and by using a higher number of patients of different ages. Finally, the CD9 upregulation in HSCs observed previously in patients affected by myeloid neoplasms [21] could be a reflection of an increase in total CD34^+^ stem cells, including VSELs, since our study is the first to differentiate the expression of CD9 between the two types of stem cells.

## Figures and Tables

**Figure 1 biology-10-00312-f001:**
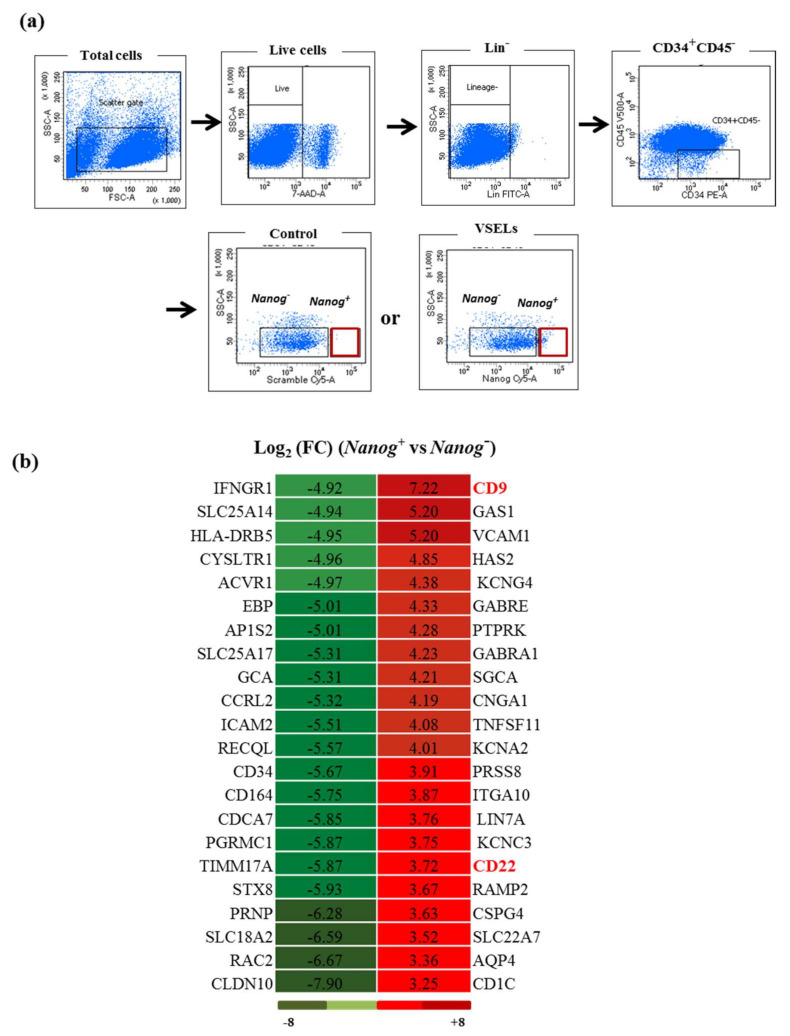
Differential expression of mRNA receptors expressed in very small embryonic-like stem cells (VSELs). (**a**) Sorting profile of the VSEL Lin^−^CD34^+^CD45^−^*Nanog^+^* and *Nanog**^−^* cells used in the RNA sequencing; the control represents cells labeled with the scramble probe. (**b**) Heat map showing the differentially expressed receptors observed by RNA-Seq. The main induced receptors in VSELs by comparison to the control cells expressed as log_2_ fold change (FC) are shown in red, and the most downregulated gene is in green (log_2_ fold change). The color scale is shown at the bottom.

**Figure 2 biology-10-00312-f002:**
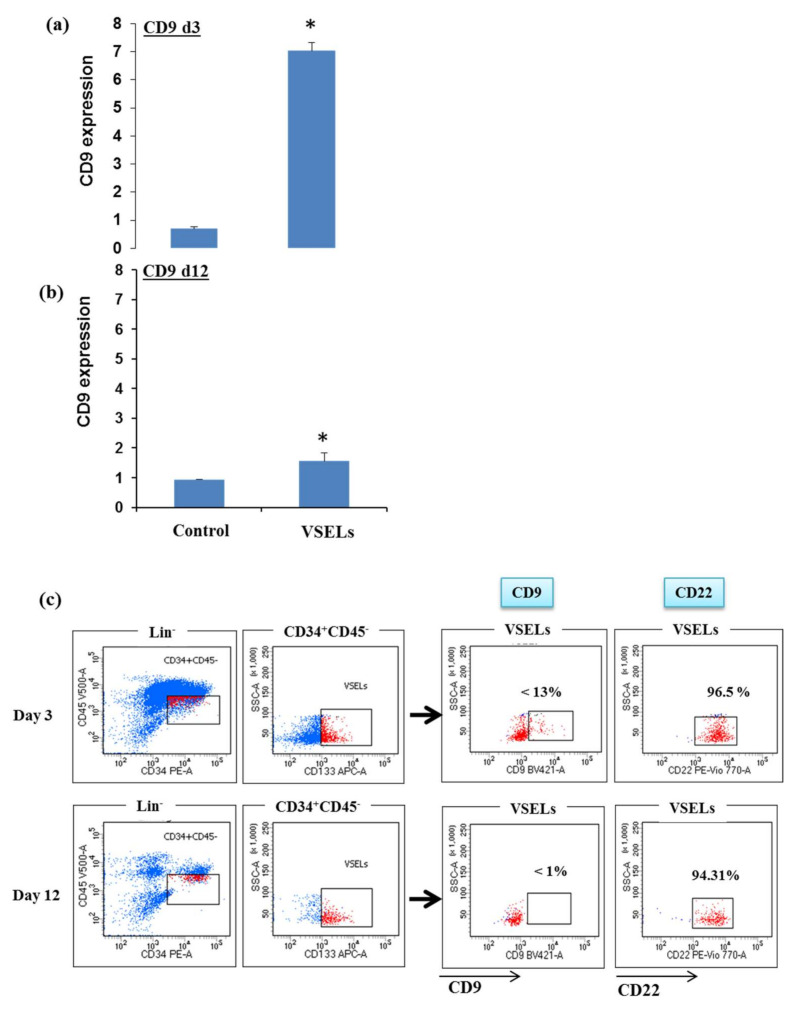
CD9 expression analysis in expanded VSELs. Relative mRNA expression analysis of CD9 in CD133^+^ VSELs and control cells, on days 3 (**a**) and 12 of expansion (**b**), were determined by real time RT-PCR. Bars represent average ± SD, *n* = 3 (* *p* < 0.05). (**c**) CD9 and CD22 surface marker expression. FACS quantification of CD9 and CD22 expression on days 3 and 12 of expansion showing low CD9 expression compared to CD22 in expanded CD133^+^ VSELs (representative experiment of *n* = 4).

**Figure 3 biology-10-00312-f003:**
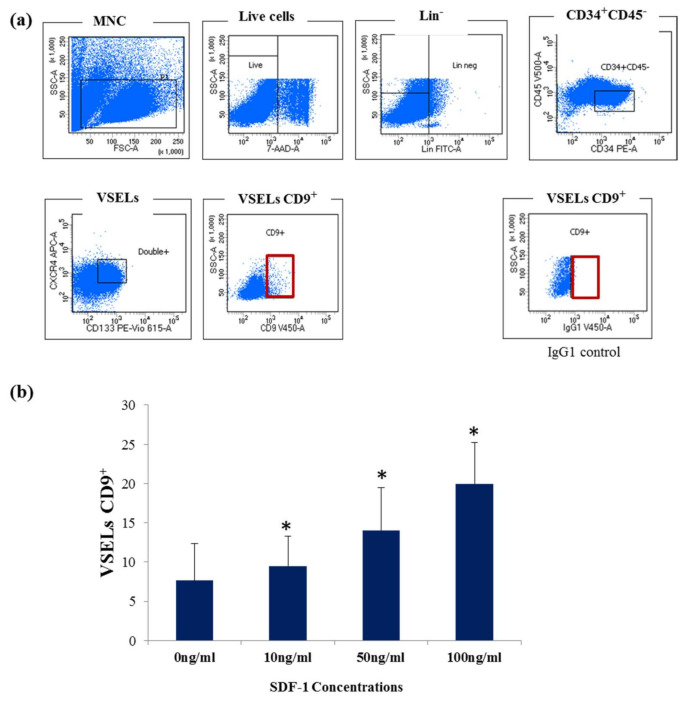
(**a**) Flow cytometry analysis of CD9 receptor expression in Lin-CD34^+^CD45^−^CD133^+^ CXCR4^+^ VSELs in the presence of SDF-1 (100 ng/mL). (**b**) The histogram represents the percentage of VSELs expressing CD9 at the indicated concentrations of SDF-1 after 4 h of incubation, represented as mean ± SD of 4 experiments (* *p* < 0.05).

**Figure 4 biology-10-00312-f004:**
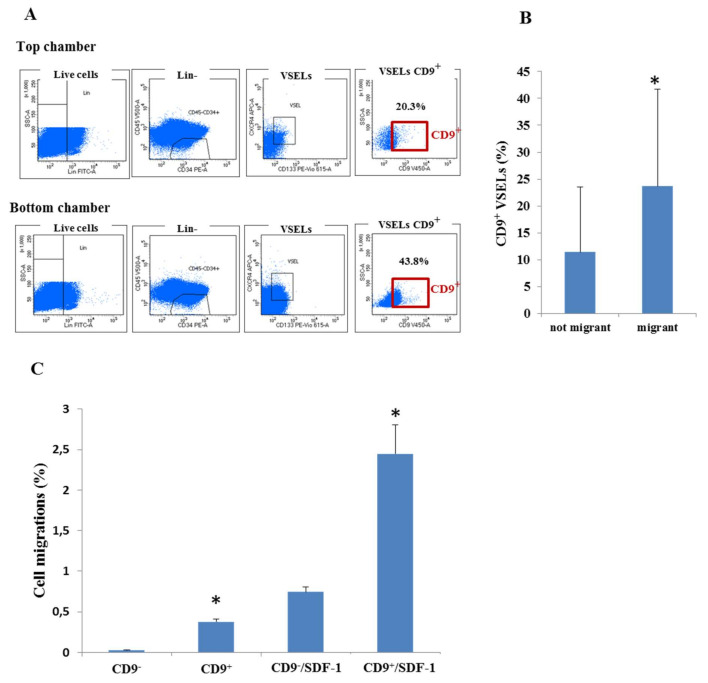
The effect of CD9 receptor expression on VSELs migration. (**A**) CD34^+^CD45^−^ cells were loaded in the upper Boyden chamber and incubated for 12 h in StemSpan medium. The presence of CD9^+^ VSELs was then determined by FACS analysis in both chambers (representative experiment). (**B**) The percentage of CD9^+^ VSELs that migrated (migrant) or not (not migrant) represented as the mean ± SD of four experiments (* *p* < 0.05). (**C**) The percentage of VSELs positive or negative for CD9^+^ that migrated in 4 h in the presence or in absence of SDF-1 (* *p* < 0.05).

**Figure 5 biology-10-00312-f005:**
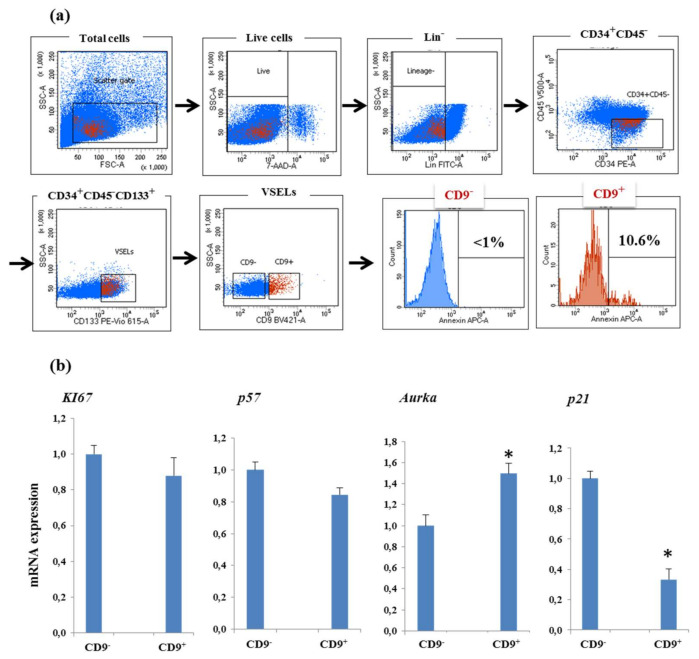
CD9 expression and VSELs survival and proliferation. (**a**) Apoptotic CD9^+^ and CD9^–^ VSEL evaluation by annexin V labeling (representative experiment). (**b**) Analysis of the expression of proliferative and apoptotic genes in VSELs positive or negative for CD9. Relative expression analysis of the indicated genes in CD133^+^ VSELs positive or negative for CD9 expression, sorted and isolated by flow cytometry and determined by real-time RT-PCR. Data are presented as means ± SD for three individual experiments (* *p* < 0.05).

**Table 1 biology-10-00312-t001:** Primer sequences.

CD9 Fw	AAGTTAGCCCTCACCATGCC
CD9 Rv	TCCAATGGCAAGGACAGCA
KI67 Fw	ATTGAACCTGCGGAAGAGCTGA
KI67 Rv	GGAGCGCAGGGATATTCCCTTA
P57 Kip2 Fw	GCGGCGATCAAGAAGCTGT
P57 Kip2 Rv	TGGCGAAGAAATCGGAGATCA
AURKA-Fw	GCATTTCAGGACCTGTTAAGGCTA
AURKA-Rv	TGCTGAGTCACGAGAACACGTTT
GAPDH Fw	CATCGCTCAGACACCATGG
GAPDH Rv	ATGTAGTTGAGGTCAATGAAGGG
β2-microglobulin Fw	TGACTTTGTCACAGCCCAAGATA
β2-microglobulin Rv	AATGCGGCATCTTCAAACCT

**Table 2 biology-10-00312-t002:** Relative CD9 receptor expression in VSELs and HSCs isolated from peripheral blood of healthy individuals and patients affected with myeloproliferative neoplasms versus lineage negatives cells.

	Controls (n = 13)	MPNs Diagnosis (n = 7)
Age (year)	37.4 ± 7	64.6 ± 17.6
WBC × 10^9^/L	3.9–10.9 ^#^	8.7 ± 6.6
RBC × 10^12^/L	4.4–5.6 ^#^	5.1 ± 112
Hemoglobin g/dL	13.5–16.9 ^#^	14.9 ± 2.9
hematocrit	40–49.4 ^#^	44.6 ± 8.6
Platelet × 10^9^/L	166–308 ^#^	529.7 ± 424.8
CD9^+^ VSELs (%)/Lin^–^	0.16 ± 0.18	1.11 ± 1.02 *
CD9^+^ HSCs (%)/Lin^–^	0.14 ± 0.1	0.17 ± 0.15

WBC, white blood cells; RBC, red blood cells; HSCs, hematopoietic stem cells; MPNs, myeloproliferative neoplasms; Lin^–^, lineage-negative cells. ^#^ The values of the controls represent the ranges observed in normal subjects. * *p* < 0.05.

## Data Availability

All data generated in this study are shown in this article, it table, and figures. Original and digitalized data are stored at IRHT and are available on request from the corresponding author.

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
