# Peer review of "Differential Expression of the Tetraspanin CD9 in Normal and Leukemic Stem Cells"

_biology, 2021, doi:10.3390/biology10040312_

Round 1
Reviewer 1 Report
Review of manuscript ID: biology-1148089 entitled ‘A Differential Expression of the Tetraspanin CD9 in Normal and Leukemic Stem Cells’ by Rachid Lahlil and colleagues
The above titled manuscript by Rachid Lahlil is a research article on CD9 which plays a crucial role in cellular growth, mobility, and signal transduction as well as in haematological malignancy. The authors observed that CD9 is among the most expressed receptors in VSELs under the steady-state but when expanded, the CD9+ VSELs decrease and are more apoptotic. CD9- VSELs had no proliferative improvement in vitro compared to those which are CD9+. Interestingly, addition of SDF-1 induces CD9 expression on VSELs surface as observed by flow cytometry and improves their migration. Furthermore, the authors observed in the phenotypically identical VSELs present in the peripheral blood of patients with myeloproliferative neoplasms compared to healthy subjects a significantly higher number of CD9+ cells.
Overall, the manuscript requires major revisions before being considered for publication.
Specific Points
- A native English-speaking person must edit the language used in the manuscript. Or the authors must seek editing services. There are many errors grammar, spelling and typographical mistakes throughout the manuscript to mention a few.
- Lines 50-51- Adult stem cells do not replace malignant leukemic cells. This is totally wrong to say the least.
- A lot of statements are difficult to understand. For example the statement ‘Interestingly, our results indicate that the percentage of VSELs 92 that are Lin-CD34+CD45- CD133+ and/or CXCR4+ positive for the CD9 receptor increases 93 significantly in a dose-dependent manner of SDF-1’ is not clear and must be made clear and concise.
- Ethics Number must be included in the manuscript
- The word ‘written’ must be included before informed consent
- The authors used Nanog positive and negative as the only criteria for isolation of VSELs. More than one marker was supposed to be used.
- Authors must make Figure 2C clear and readable on the x-axis.
- Authors must explain the use of different concentrations of SDF-1 (10, 50 and 100 ng/ml) as given in the manuscript.
- Figure 3B does not show significant differences as claimed by the authors. Authors must check their calculations.
- Figure 4 B does not show significant differences as claimed by authors. Authors must check their calculations.
- If CD9 seems to be associated with cancer stem cell properties, then that is not good. Cancer stem cells cause cancer. The authors mix cancer stem cell properties and regenerative medicine application of adult stem cells making the reader to be confused. It is best to talk about adult stem cells only and regenerative medicine and leave out the cancer stem cell part.
Author Response
Please see the attachement

Reviewer 2 Report
The manuscript of Lahlil et al. Has an interesting and possible important topic. However, several problems arise. Most of the experiments show simple statistical correlations.
- E.g. in paragraph 3.3 it should be stated that CD9 may be a surrogate marker for some biological program initiated by SDF1. With the here presented statistical correlations no real function or biological connection can be identified.
- For paragraph 3.4 and figure 5 certainly protein expression would be more informative. Importantly it should be stated that there is no functional evidence displayed. CD9 here may very well be simply a surrogate marker, e.g. for differentiation
- I see problems with the comparison of MPNs with controls. E.g. in table 2. For one the matching is terrible, since the age is very very different. Is the CD9+ VSELs (%)/Lin- population age dependent? Next MPNs are very different diseases e.g. BCR-ABL1 pos or JAK2 many of the summarized comparisons here do not make much sense since the different MPNs are so different. What was the distributions of the different MPNs? Next the range of measured values for the CD9+ VSELs (%)/Lin- population in the MPNs is huge! Even in remission! What test leads to the significant p-values? Additionally, in the paragraph 3.5 the authors reference papers describing a diagnostic role of CD9 in lymphoid diseases; but the following analysis are performed in myeloid entities; this at least calls for an explanation of why the authors think that in this case they are biologically comparable. Also the conclusion in this paragraph is not really backed by any of the observations.
- for all experiments positive controls should be shown. Eg. for figure1: Nanog
- Most of the experiments show some statistical correlations. Certainly, it would be nice to have some functional data to demonstrate CD9s function; e.g. gain/lost experiments or rescues. This at least should be stated. In my view the strong conclusions in the last paragraph of the manuscript - as it is now - is not justified.
Minor:
- a comma is missing in the author list
-the abbreviation VSLE in line 16 is confusing since its not explained (or written out); however, one expects an abbrivation here
- in line 180 BCR-ABL1, CALR, MPL, and JAK2 should be italicized; whenever RNA or DNA is shown the gen should be italicized.
- in line 181 I am not sure what a “typical” stage would be?
- in e.g. table 2 and all over the manuscript “,” should be replaced by “.” In the numbers.
Author Response
Please see the attachement.

Reviewer 3 Report
In the paper Lahlil et al. present new data on expression of tertaspanin CD9 in normal and leukemic cells. They also observed that CD9 is highly expressed in VSELs, however its expression decreases with expansion when VSELs become more apoptotic. This data suggests that CD9 is important for maintain VSELs viability. Another interesting observation is positive effect of SDF-1 on CD9 expression and VSELs viability.
Overall, this paper is interesting and provides new data on a role of CD9 in maintaining viability of these early development stem cells. It also provides new information on VSELs number in peripheral blood from patients with myeloproliferative neoplasms (MPN) at diagnosis and remission (Table 2).
Minor comments
- It would be important to characterize better MPN patients isolated VSELs. Do these cells express bcr-abl or JAK2 (V617F) transcripts.
- Since VSELs were successfully expanded by another team in a presence of FSH + LH – it would be interesting in the future to see if addition of these two hormones would increase similarly as SDF-1, expression of CD9 on VSELs.
- This paper should be read before publication by a native speaker.
Author Response
Please see the attachement.

Round 2
Reviewer 1 Report
The authors addressed all my concerns
Author Response
Dear Reviewer 1,
Thank you for your consideration,
Reviewer 2 Report
In my view the manuscript has not significantly improved. The addition of knowledge to the field is very limited. CD9 in the context described here may be a mere bystander with statistical association. However, throughout the manuscript a functional context is implied. Gain and loss of function experiments should back the statements.
The authors show - mostly in umbilical cells - that CD9 reduces when VSEL cells are cultured and that SDF1 may mitigate this effect (which was previously shown by Broska et al. was it not?). CD9 expression associates with migration of umbilical VSEL cells and with tendency towards apoptosis. This may be interesting, biologically and clinically, but here only associations are shown not functional of an actual biological connection is demonstrated; Thus the actual results are rather limited in the conclusion that may be drawn.
The implied connection to the pathogenesis of JAK2 positive MPNs is more then doubtful. The sentence in lines 297 -299 is not backed by anything and its not even suggestive… In line 180 I do not understand what a primary form of a JAK2 mutation positive MPN is (de novo vs secondary?). In the response to the reviewer the authors state that “The distribution of the MPNs included patients was 3 BCR-ABL and 13 JAK2 (V617F).” However, the text as it is now implies that all were JAK2 mutation positiv?
Whenever a gene or cDNA is mentioned, the gene should be italicized..
Author Response
Please see the attachement
